# UHFUS: A Valuable Tool in Evaluating Exocrine Gland Abnormalities in Sjögren’s Disease

**DOI:** 10.3390/diagnostics13172771

**Published:** 2023-08-26

**Authors:** Giovanni Fulvio, Rossana Izzetti, Giacomo Aringhieri, Valentina Donati, Francesco Ferro, Giovanna Gabbriellini, Marta Mosca, Chiara Baldini

**Affiliations:** 1Rheumatology Unit, Department of Clinical and Experimental Medicine, University of Pisa, 56126 Pisa, Italy; 2Department of Clinical and Translational Science, University of Pisa, 56126 Pisa, Italy; 3Unit of Dentistry and Oral Surgery, Department of Surgical, Medical and Molecular Pathology and Critical Care Medicine, University of Pisa, 56126 Pisa, Italy; 4Academic Radiology, Department of Clinical and Translational Research, University of Pisa, 56126 Pisa, Italy; 5Unit of Pathological Anatomy 2, Department of Laboratory Medicine, University of Pisa, 56126 Pisa, Italy; 6Ophthalmology, Department of Surgical, Medical and Molecular Pathology and Critical Care Medicine, University of Pisa, 56126 Pisa, Italy

**Keywords:** ultra-high frequency ultrasound, Sjögren’s disease (SjD), exocrine glands, lacrimal glands, labial salivary glands, minor salivary glands, major salivary glands, major salivary glands ultrasonography

## Abstract

Sjögren’s Disease (SjD) is a chronic autoimmune disorder that affects the salivary and lacrimal glands, leading to xerostomia and xerophthalmia. Ultrasonography of Major Salivary Glands (SGUS) is a well-established tool for the identification of the salivary glands’ abnormalities in SjD. Recently, a growing interest has arisen in the assessment of the other exocrine glands with ultrasonography: lacrimal glands (LGUS) and labial salivary glands (LSGUS). The objective of this study is to explore the practical applications of ultra-high frequency ultrasound (UHFUS) in the assessment of lacrimal glands and labial salivary glands. Indeed, UHFUS, with its improved spatial resolution compared to conventional ultrasonography, allows for the evaluation of microscopic structures and has been successfully applied in various medical fields. In lacrimal glands, conventional high-frequency ultrasound (HFUS) can detect characteristic inflammatory changes, atrophic alterations, blood flow patterns, and neoplastic lesions associated with SjD. However, sometimes it is challenging to identify lacrimal glands characteristics, thus making UHFUS a promising tool. Regarding labial salivary glands, limited research is available with conventional HFUS, but UHFUS proves to be a good tool to evaluate glandular inhomogeneity and to guide labial salivary glands biopsy. The comprehensive understanding of organ involvement facilitated by UHFUS may significantly improve the management of SjD patients.

## 1. Introduction

Sjögren’s Disease (SjD) is a chronic autoimmune disorder characterized by inflammation and dysfunction of the exocrine glands, particularly the salivary and lacrimal glands. These glands are responsible for producing saliva and tears, respectively, and their impairment in SjD leads to dryness of the mouth and eyes, known as xerostomia and xerophthalmia, respectively [1]. A valuable tool in evaluating exocrine gland abnormalities is ultrasonography of the major salivary glands (SGUS), largely employed for clinical and research purposes since it allows for the assessment of gland size, morphology, stiffness, and vascularity [2,3,4]. In SjD, SGUS can detect characteristic elementary findings such as hypoechoic areas, dilated ducts, and changes in blood flow patterns [5,6,7]. By ultrasonography, clinicians may identify the disease phenotype, disease activity, glandular damage, and prognostic information [8,9,10]. In addition, SGUS plays an important role in the detection of major salivary gland lymphoma, a possible complication of SjD [11]. In recent years, great interest has arisen in the application of ultrasonography for the diagnosis and phenotypic stratification of SjD, expanding beyond the conventional ultrasonography of major salivary glands to the ultrasonography of the other exocrine glands: lacrimal glands (LGUS) and labial salivary glands (LSGUS). Particularly, from this perspective, great potential has been seen in ultra-high frequency ultrasound (UHFUS). This study aims to provide insights into the practical applications and role of ultra-high frequency ultrasound in evaluating lacrimal glands and labial salivary glands.

## 2. Ultra-High Frequency Ultrasound

Ultra-high frequency ultrasound is a diagnostic technique that uses ultrasound frequencies higher than 30 MHz, providing improved spatial resolution compared to conventional ultrasonography. It was first introduced in the preclinical setting in the mid-90s and in the clinical setting at the beginning of the 2000s. UHFUS allows for the evaluation of anatomical structures with submillimeter resolution, making it useful in dermatology, angiology, intraoral pathology, pediatric imaging, peripheral nerve evaluation, and musculoskeletal disorders. It offers the ability to assess microscopic structures such as cutaneous and vessel layers, nerve anatomy, and lymph node structures, which are not easily visualized with conventional high-frequency ultrasound (HFUS). UHFUS has the potential to aid in the diagnosis, surgical planning, follow-up of different pathologies, and assessment of histoanatomy. However, it has a lower penetration depth compared to HFUS, and its use requires a deeper understanding of microscopic anatomy for accurate differential diagnosis. The safety of UHFUS has been studied in preclinical research, showing no significant adverse effects, but the increased frequency does raise concerns about thermal and mechanical energy deposition in human tissues. To ensure patient safety, UHFUS equipment includes real-time thermal and mechanical indices to monitor risks during the examination. Moreover, UHFUS equipment is designed to automatically adjust its output to ensure that mechanical and thermal limits are not exceeded for all imaging modalities. Overall, UHFUS offers versatility, cost-effectiveness, and non-invasiveness, making it a promising imaging modality that is continually evolving and expanding its applications in clinical practice [12,13,14].

## 3. Anatomy of Lacrimal Glands and Labial Salivary Glands

The lacrimal gland is anatomically divided into two distinct lobes: the orbital lobe and the palpebral lobe. The orbital lobe is larger, comprising approximately 76.6% of the entire gland’s weight, while the palpebral lobe accounts for the remaining 23.4%. Lacrimal gland dimensions may vary among individuals: the long axis of the orbital lobe ranges from 20 to 25 mm, its short axis from 10 to 14 mm, and its thickness from 3 to 6 mm. Instead, the long axis of the palpebral lobe ranges from 9 to 15 mm, its short axis is approximately 8 mm, and its thickness is 2 mm. The lacrimal gland’s total dimensions include a long axis of 15–20 mm, a short axis of 10–12 mm, and a thickness of 5 mm. The space between the orbital wall and the globe accommodates the orbital lobe. The palpebral lobe extends anteriorly beyond the superior orbital margin, allowing its inferior surface to contact the lateral portion of the superior fornix where its ducts open. In terms of anatomical relationships, the orbital lobe is above the elevator palpebrae superioris aponeurosis, while the palpebral lobe lies beneath it. Overall, the lacrimal gland is a bilobed serous gland situated in the superolateral aspect of the orbit, with the orbital lobe being larger and situated in the lacrimal fossa, while the palpebral lobe lies below. The lacrimal gland’s relationship with the elevator palpebrae superioris aponeurosis changes depending on eyelid movement. When the eyelids are closed, the narrower and more proximal region of the elevator palpebrae aponeurosis lies between the orbital lobe above and the palpebral lobe below that, in this position, is well detectable with ultrasonography. Conversely, during eyelid retraction, the wider anterior part of the aponeurosis moves up and back, displacing the gland in the supratemporal region of the orbit. It is worth noting that the size of the lacrimal gland might vary in individuals, particularly in older individuals, where atrophy can lead to a smaller footprint on the globe, thus making it challenging to identify with ultrasonography [15,16,17].

There are more than 1000 minor salivary glands that can be found in various anatomical regions, including the sinonasal cavity, oral cavity, pharynx, larynx, trachea, lungs, and middle ear cavity, although they are most densely concentrated in specific areas such as the buccal mucosa (inner lining of the cheeks), labial mucosa (inner lining of the lips), lingual mucosa (underside of the tongue), soft and hard palate (roof of the mouth), and floor of the mouth. The labial salivary glands are located between the mucosal epithelium of the lips and the orbicularis oris muscle and are differently distributed between the two lips: in the upper lip glands they are densely situated between the corners of the mouth, conversely, in the inferior lip they are densely situated outside the corners [18].

## 4. Ultra-High Frequency Ultrasound Scanning Technique

In our centre, Ultra-high frequency ultrasound is currently performed with Vevo MD (Visual Sonics, Toronto, ON, Canada). Regarding lacrimal glands, a linear scanner UHF48 (20–46 MHz, axial resolution 50 µm) is employed and gel is applied on the probe to allow the transmission of ultrasound. The subjects undergoing the examination are positioned in a supine (lying face up) position with their eyelids closed. The first scan is obtained placing the probe perpendicular to the skin in the upper outer space between the ocular globe and the orbit (Figure 1). Unlike HFUS, the small size of the probe allows for another scan: tilting the transducer obliquely and pointing upwards and outwards, a greater view of the gland may be achieved (Figure 2c,d). B-mode, Doppler mode, and Spectral Doppler of the lacrimal artery are acquired [19].

Regarding labial salivary glands, subjects are examined with a linear scanner UHF70 (29–71 MHz, axial resolution 30 µm) in a supine position with their neck slightly extended, the mouth subtly open, and the lower lip gently stretched. First, the transducer is adequately cleaned and disinfected. Subsequently, the gel is placed on the probe, which is enveloped with a disposable probe cover, to avoid cross-infection during an intraoral UHFUS scan. Finally, the labial mucosa (intraoral part of the lip) is scanned in successive order: the central compartment, the right compartment, and the left compartment. B-mode, Doppler mode, and Spectral Doppler of lip small vessels are acquired (Figure 3 and Figure 4) [20].

## 5. Ultrasound of Lacrimal Glands and Labial Salivary Glands

Lacrimal gland ultrasonography (LGUS) is a non-invasive imaging technique used to evaluate the lacrimal glands with several modalities: B-Scale, Doppler, Spectral Doppler, and Shear Wave Elastography (SWE). Table 1 summarizes the most recent and relevant literature studies on LGUS ultrasonography. In greyscale, lacrimal glands of patients with primary Sjögren’s Disease may exhibit specific characteristics compared to healthy individuals. These include hypoechoic areas/inhomogeneity with enlargement, suggestive of an inflammatory phase, as well as atrophic changes such as hyperechoic bands, fibrotic changes, or fatty infiltration [19,21,22]. Doppler imaging of lacrimal glands mainly allows for the assessment of high blood flow: in SjD patients, a higher percentage of intraglandular branches of lacrimal artery have been found with a higher Resistivity Index [22,23]. Figure 1 and Figure 2 show normal lacrimal glands and lacrimal glands with inhomogeneity using UHFUS and HFUS. Shear Wave Elastography is an increasingly utilized technique in lacrimal gland ultrasonography to evaluate the elasticity or stiffness of the lacrimal gland. Lacrimal glands often exhibit elevated SWE values, which suggest the presence of fibrotic changes. SWE parameters are associated with several clinical features, including OSDI (Ocular Surface Disease Index), ESSPRI (EULAR Sjögren’s Syndrome Patient Reported Index), and the occurrence of dry eye [24,25,26]. Studies have also demonstrated the utility of ultrasonography in differentiating neoplastic lesions within the lacrimal gland, such as lymphoma or non-epithelial lesions. These lesions typically appear as hypoechoic areas with central and peripheral vascularity [21,27]. Therefore, lacrimal gland ultrasonography can aid in the identification of suspicious lesions for further evaluation. In summary, lacrimal gland ultrasonography can discriminate SjD patients from healthy control, may identify lesions suspicious for lymphoma, and is associated with local disease activity and functional test. It is worth noting that lacrimal glands are located superficially, even if there are currently no studies available on the use of ultra-high frequency ultrasound. Interestingly, ultra-high frequency ultrasound has been employed to assess other eye structures, such as the lacrimal drainage system, highlighting its importance and safety in evaluating ocular pathologies [28,29,30,31,32]. In addition, with conventional ultrasound, lacrimal glands can be frequently undetectable. Conversely, UHFUS, due to a higher resolution and the possibility of performing larger gland scans, may identify small and atrophic glands.

In contrast to lacrimal glands, there is limited research available regarding conventional ultrasonography of labial salivary glands (Table 1). To date, only one study has focused on minor salivary glands, revealing how SWE values in patients with SjD are associated with disease activity measured by the ESSDAI (EULAR Sjögren’s Syndrome Disease Activity Index), levels of IgG antibodies, and the presence of hypocomplementemia [33]. Research conducted by our group has yielded insights into the utility of ultra-high frequency ultrasound in assessing labial salivary glands [34]. Figure 3 and Figure 4 show normal LSG and LSG with inhomogeneity and a cystic lesion using UHFUS and HFUS, respectively. UHFUS showed a good reliability to assess glandular inhomogeneity that in turn was associated with histological inflammation and serology. More specifically, in a cross-sectional study including 128 patients with suspected SjD, we found that UHFUS was able to discriminate SjD from no-SjD sicca controls, as LSG inhomogeneity was significantly higher in patients with SjD than in no-SS subjects [35,36]. The study highlighted the optimal feasibility of UHFUS and its high sensitivity in identifying negative patients on subsequent lip biopsy. Interestingly, our preliminary data indicate that UHFUS exhibits a specificity for SjD diagnosis from approximately 65% to 85% depending on the chosen cut-off threshold [35,37]. Furthermore, findings suggestive of lymphoproliferative lesion have been identified; there being very hypoechoic areas and high Doppler signal that were detected in a patient with a more complex inflammatory infiltrate in her biopsy [38]. Eventually, UHFUS may be useful to guide biopsy and to improve sampling of labial salivary glands [20].

**Table 1 diagnostics-13-02771-t001:** Major studies on conventional ultrasound of lacrimal gland and labial salivary glands.

Author (Year)	US Modalities	Main Findings
Lacrimal Gland Conventional Ultrasonography		
Giovagnorio et al. (2000) [21]	B-mode Colour Doppler Spectral Doppler	When well visibile, lacrimal glands in SjD patients are enlarged and hypoechoic Presence of Hyperechoic bands in SjD patients Two lymphoma identified with cyst-like lesions RI higher than normal individuals
Bilgili et al. (2004) [23]	Spectral Doppler	Values of RI and PI in normal Lacrimal Artery
De Lucia et al. (2020) [19]	B-mode Colour Doppler	SjD patients have higher proportion of inhomogeneity and fibrous gland appearence
Kim et al. (2022) [22]	B-mode Colour Doppler	SjD patients have higher proportion of intraglandular branch of lacrimal artery, inhomogeneity, hyperechoic bands SjD diagnostic value of intraglandular branch and inhomogeneity
Świecka et al. (2023) [24]	Shear Wave Elastography	SjD diagnostic value of SWE (SjD patients have higher SWE values)
Karadeniz et al. (2023) [25]	Shear Wave Elastography	SjD diagnostic value of SWE (SjD patients have higher SWE values) Correlation of SWE with OSDI and ESSPRI
Yılmaz et al. (2023) [26]	Shear Wave Elastography	SWE values are higher in patients with dry eye
Labial salivary Gland Conventional Ultrasonography		
Wang et al. (2022) [33]	Shear Wave Elastography	SWE values are associated with ESSDAI, IgG values and hypocomplementemia
Labial salivary Gland ultra-high frequency Ultrasonography		
Ferro et al. (2020) [35]	B-mode	SjD diagnostic value (SjD patients have higher inhomogeneity) Associations of inhomogeneity with Ro/SSA+ positivity Correlationsof inhomogenity with histological inflammation
Izzetti et al. (2021) [20]	B-mode	Support to the biopsy procedure

SjD = Sjögren’s Disease, RI = Resistivity Index, PI = Pulsatility Index, SWE = Shear Wave Elastography, OSDI = Ocular Surface Disease Index, ESSPRI = EULAR Sjogren’s Syndrome Patient Reported Index, ESSDAI = EULAR Sjögren’s syndrome disease activity index.

## 6. Conclusions

Ultra-high frequency ultrasound (UHFUS) is a versatile and non-invasive imaging modality that is increasingly employed in patients with Sjögren’s Disease for research and clinical purposes. UHFUS, indeed, has the potential to better characterize both normal and pathological findings in the exocrine glands, including lacrimal glands and labial salivary glands. Furthermore, due to its enhanced resolution and associations with microanatomy, UHFUS may contribute to a more comprehensive understanding of glandular abnormalities in SjD. Future biopsy-based prospective clinical studies including a larger patient cohort are warranted to comprehensively define the potential role of ultra-high frequency ultrasound (UHFUS) in both the diagnosis of Sjögren’s Disease (SjD) and the phenotyping of affected patients. The comparison between UHFUS findings and histology appears instrumental to better define the correspondence between sonographic and histological biomarkers. Indeed, a comprehensive understanding of glandular involvement in SjD may improve diagnosis, management, and treatment in patients with SjD.

## Figures and Tables

**Figure 1 diagnostics-13-02771-f001:**
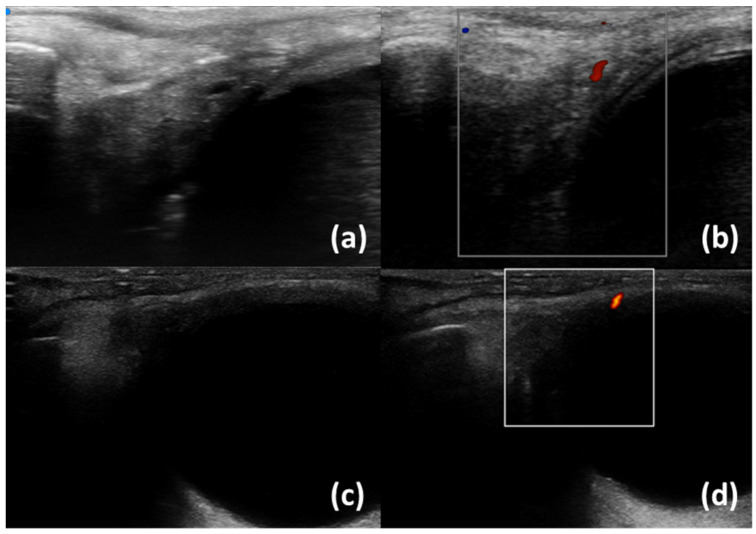
Normal lacrimal glands with a branch of the lacrimal artery. UHFUS (**a**,**b**) and HFUS (**c**,**d**); B-scale (**a**,**c**) and Colour Doppler (**b**,**d**).

**Figure 2 diagnostics-13-02771-f002:**
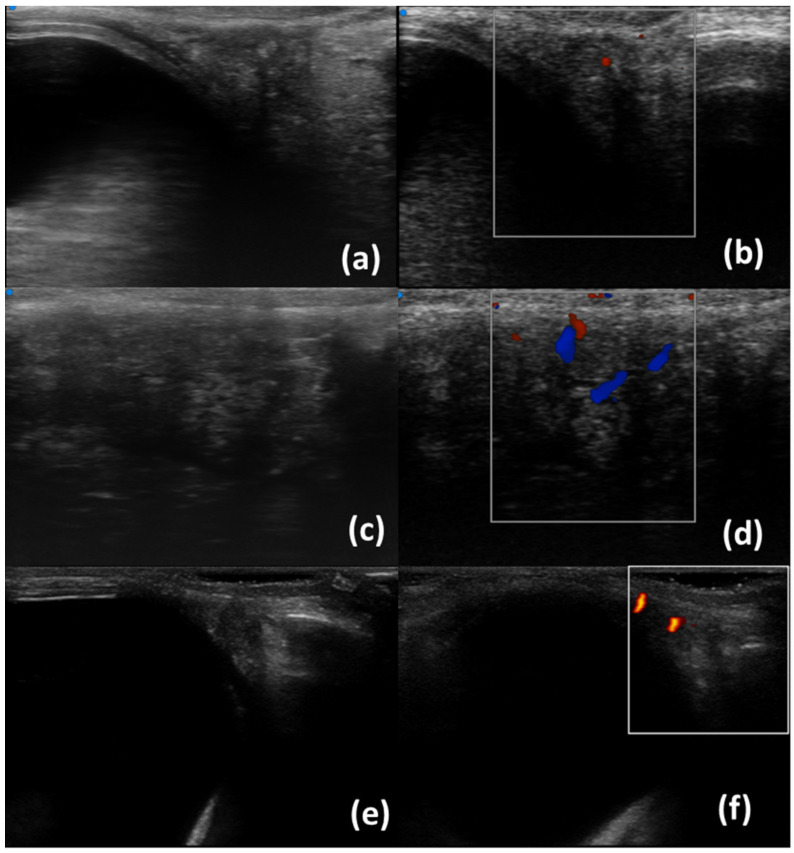
Lacrimal glands with inhomogeneity. UHFUS (**a**–**d**) and HFUS (**c**,**d**); B-scale (**a**,**c**,**f**) and Colour Doppler (**b**,**d**,**e**). Scans in figures (**c**,**d**) were obtained by tilting the probe upwards and outwards.

**Figure 3 diagnostics-13-02771-f003:**
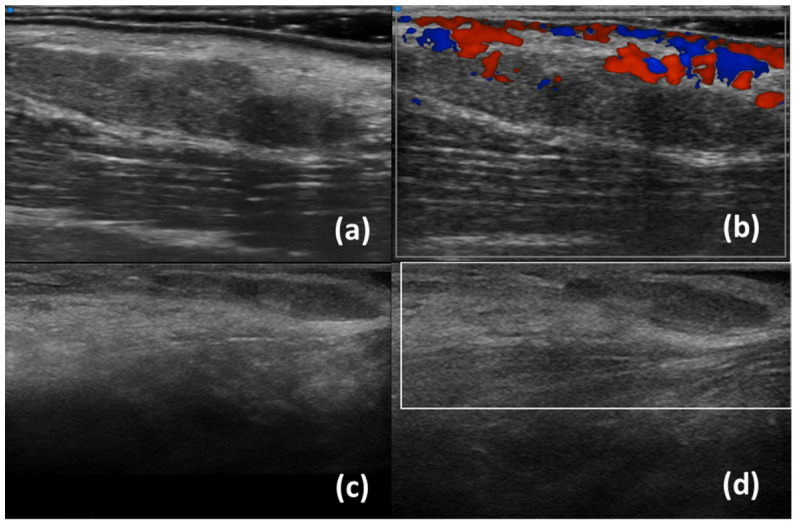
Normal labial salivary glands using UHFUS (**a**,**b**) and HFUS (**c**,**d**); B-scale (**a**,**c**) and Colour Doppler (**b**,**d**).

**Figure 4 diagnostics-13-02771-f004:**
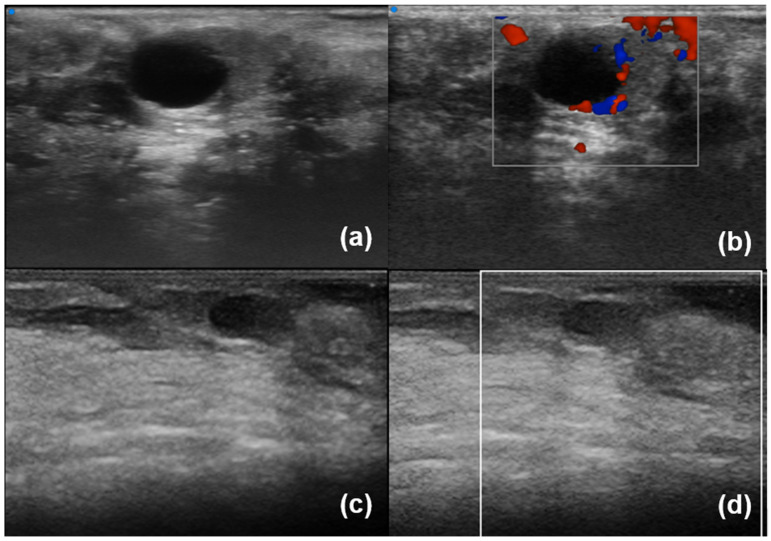
Labial salivary glands with inhomogeneity and a cystic lesion. UHFUS (**a**,**b**) and HFUS (**c**,**d**); B-scale (**a**,**c**) and Colour Doppler (**b**,**d**).

## Data Availability

Data is contained within the article.

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
