# Peer review of "UHFUS: A Valuable Tool in Evaluating Exocrine Gland Abnormalities in Sjögren’s Disease"

_diagnostics, 2023, doi:10.3390/diagnostics13172771_

Round 1
Reviewer 1 Report
This authors well described the use of ultra-high frequency ultrasound (UHFUS) in Sjögren’s disease. In fact it is kind of narrative review paper how to apply UHFUS in this disease and the potential role of UHFUS in the diagnosis and phenotyping in Sjögren’s disease is (the authors use both Sjögren’s disease (the newer name) and Sjögren’s syndrome (the ‘old’ name); use just one name of this condition). Clearly, healthy controls and Sjögren’s disease can be distinguished, but how specific is UHFUS. This has to be discussed as well.
Furthermore, I do not understand why the authors mention: “Institutional Review Board Statement: The study was conducted in accordance with the Declara[1]tion of Helsinki, and approved by the Institutional Review Board of the University Hospital of Pisa (Comitato Etico Area Vasta Nord-Ovest, CEAVNO) (protocol code 14540 approved on 14 March 2019). Informed Consent Statement: Informed consent was obtained from all subjects involved in the study.” That is not needed as no patients are involved in this narrative type of review.
Just a minor typo 128 patientients
Author Response
Thank you for your valuable feedback.
We have updated the manuscript to use the term "Sjögren’s Disease" consistently throughout the paper and we added the specificity, as per your suggestion. We agree that being this a narrative review and not a clinical study involving patients information about the IRB approval could be removed; however, please note that the images included in our paper were collected from patients participating in an observational trial that we are performing in our Unit that has been approved by the IRB. As regards minor typo, the correction has been made, and the mistake has been rectified in the revised version of the paper.
Reviewer 2 Report
The manuscript is interesting, however some things can be improved:
1.two many key words identical with those in a title; it should be changed.
2. "Future studies are warranted to define the role of UHFUS in the diagnosis of SjD and in patients’ phenotyping". -it needs more details: what kind of study, how many patients etc.
3."Minor salivary glands are about 1000 and..... "-something is missing in this sentence.
Author Response
Thank you for your valuable feedback.
We have reviewed the keywords to address the issue of similarity. In addition, we have revised the section regarding future studies to provide more information, including the type of studies and the number of the patients. Finally, to complete the sentence, we have also specified that we considered the overall number of minor salivary glands of the human body.